# A FORTRAN Program to Model Magnetic Gradient Tensor at High Susceptibility Using Contraction Integral Equation Method

**Longwei Chen** [1] and **Fang Ouyang** [2,*]

1   Guangxi Key Laboratory of Exploration for Hidden Metallic Ore Deposits, College of Earth Sciences, Guilin University of Technology, Guilin 541006, China; longweichen_glut@glut.edu.cn
2   College of Geophysics, China University of Petroleum-Beijing, Beijing 102249, China
*   Correspondence: fangouyang92@163.com

**Abstract:** The magnetic gradient tensor provides a powerful tool for detecting magnetic bodies because of its ability to emphasize detailed features of the magnetic anomalies. To interpret field measurements obtained by magnetic gradiometry, the forward calculation of magnetic gradient fields is always necessary. In this paper, we present a contraction integral equation method to simulate the gradient fields produced by 3-D magnetic bodies of arbitrary shapes and high susceptibilities. The method employs rectangular prisms to approximate the source region with the assumption that the magnetization in each element is homogeneous. The gradient fields are first solved in the Fourier domain and then transformed into the spatial domain by 2-D Gauss-FFT. This calculation is performed iteratively until the required accuracy is reached. The convergence of the iterative procedure is ensured by a contraction operator. To facilitate application, we introduce a FORTRAN program to implement the algorithm. This program is intended for users who show interests in 3D magnetic modeling at high susceptibility. The performance of the program, including its computational accuracy, efficiency and convergence behavior, is tested by several models. Numerical results show that the code is computationally accurate and efficient, and performs well at a wide range of magnetic susceptibilities from 0 SI to 1000 SI. This work, therefore, provides a significant tool for 3D forward modeling of magnetic gradient fields at high susceptibility.

**Keywords:** magnetic gradient tensor; high susceptibility; contraction integral equation; magnetic forward modeling

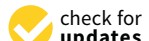

## 1. Introduction

Magnetic surveying is a significant and widely used geophysical exploration technique. Using magnetic measurements, three kinds of data can be obtained: total magnetic intensity (TMI) data, three-components field data and full tensor magnetic gradient data [1]. Because of easy acquisition, the TMI data and the three-components data were most commonly measured in the past decades [2–5]. However, these measurements are seriously affected by the Earth's background magnetic field and very sensitive to instrument orientation, hence not just functions of the target's magnetic susceptibility [6]. Recently, full tensor magnetic gradient measurements become available with the development of SQUID-based sensors [1,4,7]. In comparison with the traditional magnetic field measurements, gradient measurements offer many potential advantages, which includes less sensitivity to instrument orientation, improved resolution of shallow targets, no necessity for base stations and diurnal corrections, and suppression of regional anomalies [4,8–10]. Mathematically, the gradient tensor is a second rank tensor with $3 \times 3 = 9$ components, which is defined by derivatives of the magnetic field vector in each of the three directions of three-dimensional space [11]. Each component of the gradient tensor represents a directional filter and hence makes certain structures and characteristics of the magnetic bodies more noticeable [10].

Therefore, the magnetic gradient tensor can work as a useful tool for detecting magnetic bodies by suppressing specific undesirable contributes and emphasizing certain features of the magnetic fields.

Magnetic forward modeling of gradient fields plays a significant role in the interpretation of field measurements obtained from magnetic gradiometry. During the past decades, various analytic and numerical methods have been developed to simulate magnetic fields. Generally, analytic methods are limited to provide closed-form field expressions for magnetic bodies with regular geometric shapes [12–15]. For magnetic anomalies with arbitrary and complex susceptibility distribution, numerical methods have to be used. The spatial-domain method is one of the most commonly used numerical methods for computing magnetic fields due to bodies of given shapes [16–22]. These methods often represent the magnetized body using a set of cells (such as cubical cells) and approximate the total field by the sum of the elementary fields [23,24]. One key limitation of the spatial-domain method is that its computational time dramatically increases with the size and complexity of the model. The frequency-domain method is another important approach to model magnetic anomalies [25–27]. This approach is performed in the frequency domain and thereby can achieve high computational efficiency by using fast Fourier transform techniques. On the whole, however, most existing numerical methods ignore the effects of self-demagnetization and are only applicable to magnetic bodies with low susceptibilities.

To overcome the problem,Ref. [28] develop an iterative method and use a segmented model consisting of spherical voxels with arbitrary diameter to calculate the magnetic fields at high susceptibilities through an iterative procedure. Ref. [29] conduct a comprehensive study of the self-demagnetization effects on magnetic data, and compare the capability of two existing inversion methods in interpretation of data from highly magnetic areas. More recently,Ref. [30] develop an efficient and accurate frequency-domain iterative method that can be used to simulate magnetic fields from magnetic bodies with arbitrary shapes in a wide range of magnetic susceptibilities (0∼1000 SI). This strategy is based on a contraction integral equation and can achieve fast convergence. Although these methods perform well at high susceptibilities, they mainly focus on the calculation of the magnetic fields, and take no account of the gradient tensor.

In this paper, we present an algorithm to compute the magnetic gradient fields produced by 3-D magnetic bodies of arbitrary shapes and high susceptibilities based on the contraction integral equation method developed by [30]. In order to simulate the magnetic gradient fields for high susceptibility and to facilitate application of the algorithm of [30], we develop a computer program using FORTRAN language. This FORTRAN program generates multi-component fields, which includes six components of the magnetic gradient tensor and three components of the magnetic field vector, and has a good performance at a wide range of magnetic susceptibilities ($0 < \chi \leq 1000$ SI), particularly applicable for strongly magnetic bodies.

The remainder of the paper is organized as follows. In Section 2, the algorithm for calculating the magnetic gradient fields from strongly magnetic bodies is developed. Subsequently, a detailed introduction is given to the subroutines, inputs and outputs of the FORTRAN program. Then, in Section 4 the performance of the algorithm is tested using several simple models, and the applicability of the code is demonstrated with a synthetic two-dike model. In Section 5, some conclusions are drawn from the work.

## 2. Contraction Integral Equation Method

The full magnetic gradient tensor **T** can be obtained by taking derivatives of the magnetic vector **B** with respect to the coordinates $x$, $y$ and $z$,

$$\mathbf{T} = \begin{pmatrix} T_{xx} & T_{xy} & T_{xz} \\ T_{yx} & T_{yy} & T_{yz} \\ T_{zx} & T_{zy} & T_{zz} \end{pmatrix} = \begin{pmatrix} \partial_x B_x & \partial_x B_y & \partial_x B_z \\ \partial_y B_x & \partial_y B_y & \partial_y B_z \\ \partial_z B_x & \partial_z B_y & \partial_z B_z \end{pmatrix} \tag{1}$$

where $\partial_x$, $\partial_y$, and $\partial_z$ denote the partial derivation with respect to $x$, $y$ and $z$, respectively. $\mathbf{B} = \mu_0\,\mathbf{H}$ and $\mu_0$ is the permeability of free space. Using

$$\mathbf{H} = -\nabla U \tag{2}$$

we can also write **T** as

$$\mathbf{T} = -\mu_0 \begin{pmatrix} \partial^2_{xx}U & \partial^2_{xy}U & \partial^2_{xz}U \\ \partial^2_{xy}U & \partial^2_{yy}U & \partial^2_{yz}U \\ \partial^2_{xz}U & \partial^2_{yz}U & \partial^2_{zz}U \end{pmatrix} \tag{3}$$

It is apparent from Equation (3) that the magnetic gradient tensor **T** is symmetric. This indicates that using six components of **T**, namely $T_{xx}$, $T_{yy}$, $T_{zz}$, $T_{xy}$, $T_{xz}$ and $T_{yz}$, will be sufficient in the forward calculation. Actually, because of $\nabla \cdot \mathbf{B} = 0$, the magnetic gradient tensor has only five independent components.

To obtain the gradient fields at high susceptibility, we will first calculate the anomalous magnetic fields using the iterative contraction integral equation method, and then derive frequency-domain expressions for the anomalous gradient tensor based on the resulting magnetic field vector in the following sections.

### 2.1. The Integral Equation

In a 3D Cartesian coordinate system with downward positive $z$ direction, the integral equation with respect to the total magnetic field **H** can be written as [30]

$$\mathbf{H}(\mathbf{r}) = \mathbf{H}^0(\mathbf{r}) + \iiint\limits_{V'} \mathbf{G}(\mathbf{r},\mathbf{r}') \cdot \left( \chi(\mathbf{r}')\mathbf{H}(\mathbf{r}') \right) dV' \tag{4}$$

where **r** and $\mathbf{r}'$ are the observation point and the source point in the source region, respectively. $\chi\mathbf{H}$ denotes the magnetization **M**. $\mathbf{G} = \nabla\nabla G$ is the Green's function tensor, $\nabla$ is the gradient operator, and $G = 1/(4\pi|\mathbf{r} - \mathbf{r}'|)$ is the scalar Green's function, which satisfies the following differential equation,

$$\nabla^2 G(\mathbf{r},\mathbf{r}') = -\delta(\mathbf{r} - \mathbf{r}') \tag{5}$$

where $\delta$ denotes the Dirac delta function [19].

The anomalous fields are defined as

$$\mathbf{H}^a(\mathbf{r}) = \iiint\limits_{V'} \mathbf{G}(\mathbf{r},\mathbf{r}') \cdot \left( \chi(\mathbf{r}')\mathbf{H}(\mathbf{r}') \right) dV' \tag{6}$$

Equation (4) indicates that the total magnetic field **H** results from two parts: the background field $\mathbf{H}^0$, which is associated with the Earth's magnetic field (EMF), and the anomalous fields $\mathbf{H}^a$, which results from magnetic bodies in the source region. In general, the anomalous fields can be neglected at low magnetic susceptibilities ($\chi < 0.01$ SI) [31]. Thus, the magnetization reduces to $\mathbf{M} = \chi\mathbf{H}^0$ in this case, and accurate solutions for the integral Equation (4) can be obtained by direct methods [32,33]. For high susceptibility, however, the induced magnetization from neighboring material significantly affects the magnetic field at any point in the medium, which consequently reduces the resultant magnetic field. This phenomenon is the so-called self-demagnetization [34]. In the case of large susceptibility, the demagnetization effect is significant and cannot be neglected. Therefore, the anomalous magnetic fields must be taken into consideration, and the integral Equation (4) has to be solved by an appropriate iterative procedure.

### 2.2. Iterative Scheme

To obtain magnetic fields at high susceptibilities, [30] developed a convergent iterative scheme that efficiently calculates accurate magnetic fields produced by strongly magnetic

bodies. Here, we follow this method to determine the Fourier-domain anomalous magnetic fields and then extend the algorithm to compute the magnetic gradient tensor.

In our algorithm, rectangular prisms are used as the building blocks. The whole computational domain is considered as the source region and is evenly divided into $N_x \times N_y \times N_z$ prisms in the $x$, $y$ and $z$ directions (see Figure 1). The geometric center of the prismatic element is $(x_l, y_m, z_n)$, and the length is $\Delta x$, $\Delta y$ and $\Delta z$ in the three directions, respectively. The magnetization is assumed to be homogeneous in each prism.

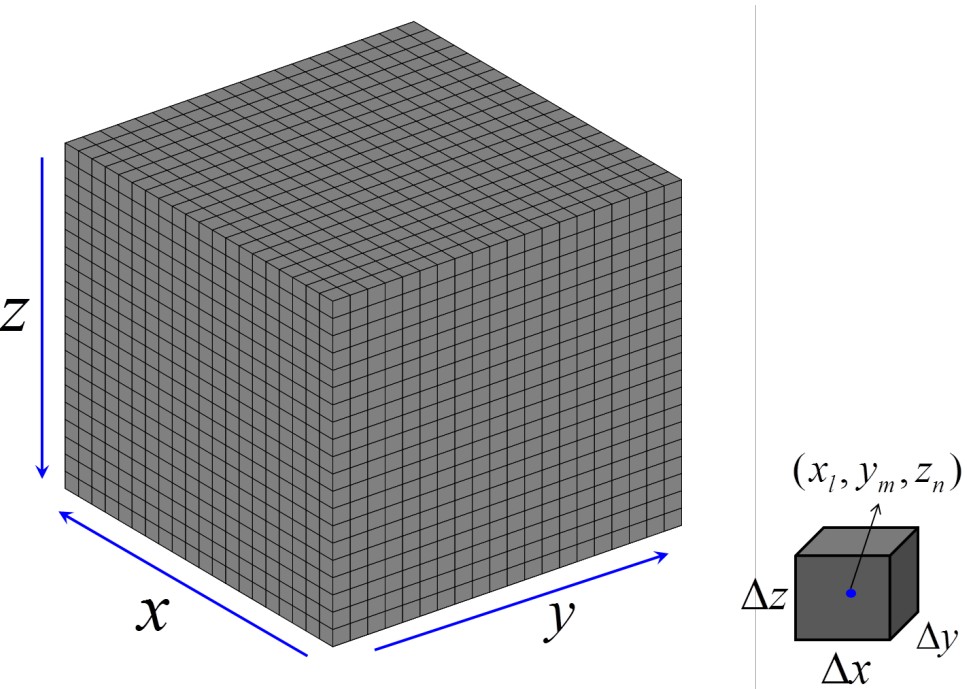

**Figure 1.** The discretization and coordinate system of the computation space. The observation points are located on the first horizontal plane and have coincident $(x, y, z)$ coordinates with the source points.

To simplify mathematical operations, we start with the anomalous magnetic potential $U^a$ [30,35],

$$U^a(\mathbf{r}) = \iiint\limits_{V'} \mathbf{G_U}(\mathbf{r}, \mathbf{r}') \cdot \mathbf{M}(\mathbf{r}') dV' \tag{7}$$

where $\mathbf{G}_U$ is the negative gradient of the scalar Green's function, i.e., $\mathbf{G}_U = -\nabla G$ and $\mathbf{M} = \chi \mathbf{H}$. Using the spatial discretization strategy aforementioned, we can get the discrete form of $U^a$

$$U^a(\mathbf{r}) = \sum_{n=1}^{N_z} \sum_{m=1}^{N_y} \sum_{l=1}^{N_x} \left( \int\limits_{z_n - 0.5\Delta z}^{z_n + 0.5\Delta z} \int\limits_{y_m - 0.5\Delta y}^{y_m + 0.5\Delta y} \int\limits_{x_l - 0.5\Delta x}^{x_l + 0.5\Delta x} \mathbf{G}_U(\mathbf{r}, \mathbf{r}') dx' dy' dz' \right) \cdot \mathbf{M}(x_l, y_m, z_n) \tag{8}$$

We deal with Equation (8) in the frequency domain of the horizontal coordinates so that the triple integral involved in the equation can reduce to a single integral only associated with $z$.

Taking the 2D Fourier transform of Equation (8) with respect to $x$ and $y$, we have

$$\hat{U}^a(z) = W \sum_{n=1}^{N_z} \mathbf{I}_G(z_n, z) \cdot \hat{\mathbf{M}}(z_n) \tag{9}$$

with

$$\mathbf{I}_G(z_n, z) = \int\limits_{z_n-0.5\Delta z}^{z_n+0.5\Delta z} \hat{\mathbf{G}}_U(z-z')\mathrm{d}z' \tag{10}$$

$$W = \frac{4\sin(0.5k_x\Delta x)\sin(0.5k_y\Delta y)}{k_xk_y\Delta x\Delta y} \tag{11}$$

where $k_x$ and $k_y$ are wavenumbers in the $x$ and $y$ directions. $\hat{U}^a$ and $\hat{\mathbf{M}}$ are the corresponding spectrum of $U^a$ and $\mathbf{M}$, respectively. According to $\mathbf{G}_U = -\nabla G$, the 2D Fourier transform of $\mathbf{G}_U$ becomes

$$\hat{\mathbf{G}}_U(z-z') = \left( -ik_x, -ik_y, sign(z-z')\cdot k \right)\frac{e^{-k|z-z'|}}{2k} \tag{12}$$

Here, $sign(z-z')$ is the sign function and $k = \sqrt{k_x^2 + k_y^2}$.

Substituting Equations (10)–(12) into Equation (9), the anomalous magnetic potential in the Fourier domain is obtained

$$\hat{U}^a(z) = \frac{W}{2k^2}\left[ e^{-kz}\sum_{z_n<z} e^{kz_n}A_1(z_n)H(z_n) \right.$$
$$+ e^{-kz}\sum_{z_n=z} e^{kz_n}A_2(z_n)H(z_n) + e^{kz}\sum_{z_n=z} e^{-kz_n}A_2(z_n)F(z_n) - 2P(z)$$
$$\left. + e^{kz}\sum_{z_n>z} e^{-kz_n}A_1(z_n)F(z_n) \right] \tag{13}$$

with

$$\begin{aligned}
A_1(z_n) &= e^{-0.5k\Delta z} - e^{0.5k\Delta z}\\
A_2(z_n) &= e^{-0.5k\Delta z}\\
P(z_n) &= ik_x\hat{M}_x(z_n) + ik_y\hat{M}_y(z_n)\\
F(z_n) &= P(z_n) + k\hat{M}_z(z_n)\\
H(z_n) &= P(z_n) - k\hat{M}_z(z_n)
\end{aligned} \tag{14}$$

where $\hat{M}_x$, $\hat{M}_y$ and $\hat{M}_z$ are three components of $\hat{\mathbf{M}}$.

Note that $P$, $F$ and $H$ in Equation (14) are functions of $\hat{\mathbf{M}}$, where $\hat{\mathbf{M}} = 2\mathrm{DFT}[\chi\mathbf{H}]$ with $2\mathrm{DFT}[\cdot]$ being a 2D Fourier transform operator. This implies that the values of $P$, $F$ and $H$ will change with the number of iterations, since the magnetic field vector $\mathbf{H}$ is renewed iteratively. In contrast, $A_1$, $A_2$ and $e^{\pm kz_n}$ are all independent of $\hat{\mathbf{M}}$ and hence can be pre-computed so as to improve the computational efficiency.

According to Equation (2), we can also write the Fourier-domain anomalous magnetic field as

$$\hat{\mathbf{H}}^a(z) = -\left( ik_x, ik_y, \partial_z \right)\hat{U}^a(z) \tag{15}$$

As can be seen, the vertical component $\hat{H}_z^a$ presents an implicit form due to the spatial derivative $\partial_z\hat{U}^a$. In order to make it explicit, we substitute Equation (13) into Equation (15) and then have

$$\hat{H}_z^a(z) = \frac{W}{2k}\left[ -e^{-kz}\sum_{z_n<z} e^{kz_n}A_1(z_n)H(z_n) \right.$$
$$-e^{-kz}\sum_{z_n=z} e^{kz_n}A_2(z_n)H(z_n) + e^{kz}\sum_{z_n=z} e^{-kz_n}A_2(z_n)F(z_n)$$
$$\left. + e^{kz}\sum_{z_n>z} e^{-kz_n}A_1(z_n)F(z_n) \right] \tag{16}$$

Therefore, once $\hat{\mathbf{H}}^a$ is determined, the 2D inverse Fourier transform of $\hat{\mathbf{H}}^a$ immediately gives its spatial-domain counterpart $\mathbf{H}^a$.

To solve $\mathbf{H}^a$, we rewrite the integral Equation (4) as

$$\mathbf{H}(\mathbf{r}) = \mathbf{H}^0(\mathbf{r}) + \mathbf{H}^a(\mathbf{r}) \tag{17}$$

From Equations (14)–(17), we can find that $\mathbf{H}^a$ is a function of $\mathbf{M}$ (or $\mathbf{H}$, since $\mathbf{M} = \chi\mathbf{H}$). Thus, a convergent iterative algorithm is necessary, in order to obtain the magnetic fields caused by strongly magnetic bodies. But the fact is that the iterative calculation of Equation (4) does not always converge, because $\ell_2$-norm of the linear integral operator in Equation (4) is bigger than 1 in general cases [30]. It indicates that a contraction operator must be used to ensure convergence stability and convergence rates.

Therefore, instead of Equation (17), we adopt the contraction integral equation developed by [30]

$$\mathbf{H}^{(j+1)}(\mathbf{r}) = \frac{2\left[\mathbf{H}^0(\mathbf{r}) + \mathbf{H}^a(\mathbf{r}, \mathbf{H}^{(j)}(\mathbf{r}))\right] + \chi\mathbf{H}^{(j)}(\mathbf{r})}{2 + \chi}, \quad j = 0, 1, 2, \cdots \tag{18}$$

By doing this, an iterative format which makes the successive calculation of the total magnetic field always convergent is established using Equations (13)–(16) and (18).

Our final purpose is to calculate the anomalous magnetic gradient tensor produced by magnetic bodies with high susceptibilities. This can be achieved by taking 2D Fourier transform of Equation (1) and making use of Equation (13). It gives the anomalous gradient tensor as

$$\hat{\mathbf{T}}^a = \mu_0 \begin{pmatrix} ik_x \hat{H}^a_x & ik_y \hat{H}^a_x & ik_x \hat{H}^a_z \\ & ik_y \hat{H}^a_y & ik_y \hat{H}^a_z \\ sym. & & -\partial^2_{zz}\hat{U}^a \end{pmatrix} \tag{19}$$

with

$$\begin{aligned} \partial^2_{zz}\hat{U}^a = \frac{W}{2}\Bigg[ & e^{-kz}\sum_{z_n < z} e^{kz_n} A_1(z_n) H(z_n) \\ & + e^{-kz}\sum_{z_n = z} e^{kz_n} A_2(z_n) H(z_n) + e^{kz}\sum_{z_n = z} e^{-kz_n} A_2(z_n) F(z_n) \\ & + e^{kz}\sum_{z_n > z} e^{-kz_n} A_1(z_n) F(z_n)\Bigg] \end{aligned} \tag{20}$$

where $\hat{\mathbf{H}}^a$ is given by Equations (15) and (16). It is noteworthy that the calculation of the gradient tensor should be carried out only when the iterative computation of the magnetic field is totally completed or the required convergence accuracy is reached. Certainly, the gradient tensor can also be calculated iteratively, but this only leads to an unnecessary computational cost.

Finally, we obtain the spatial-domain gradient fields through 2D inverse Fourier transform.

### 2.3. Workflow for Modeling Gradient Fields

The workflow for modeling magnetic gradient tensor at high susceptibility is summarized in Figure 2. In this study, the Gauss-FFT technique with 4 nodes is applied, and the root mean square (rms) difference is adopted as the convergence criterion, that is,

$$\text{rms} = \sqrt{\frac{\sum_{i,m,n}\left|\mathbf{H}^{(j)}(x_i, y_m, z_n) - \mathbf{H}^{(j+1)}(x_i, y_m, z_n)\right|^2}{N_x N_y N_z}} \tag{21}$$

where $\mathbf{H}^{(j)}$ denotes the total magnetic field after the *j*-th iteration.

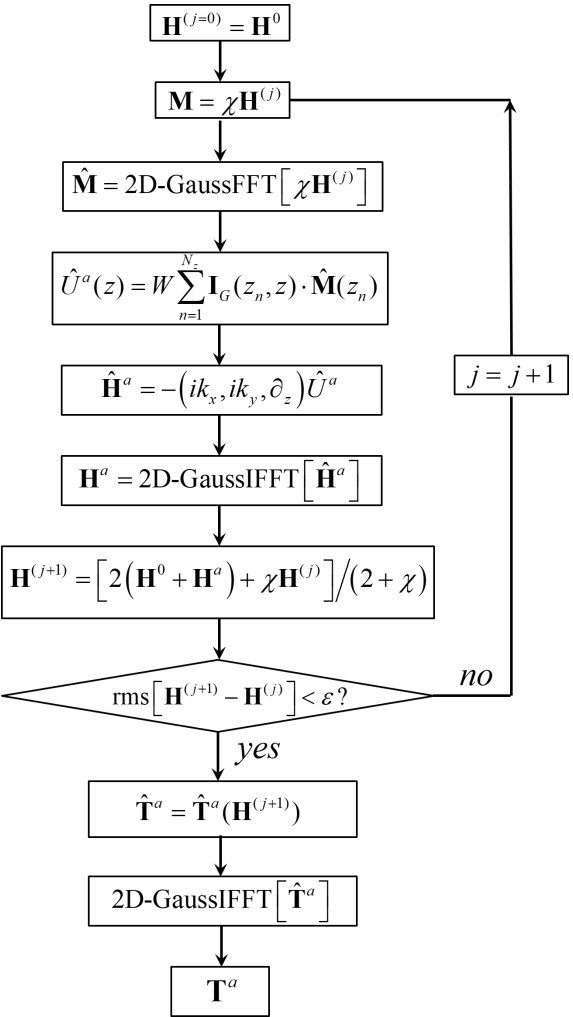

**Figure 2.** Workflow for modeling the magnetic gradient tensor produced by 3D magnetic bodies with high susceptibility (modified after [30]).

## 3. Description of The Code

### 3.1. Subroutines

The developed FORTRAN program for calculating magnetic gradient fields at high susceptibilities consists of a main program, six major subroutines and four supporting subroutines. The functions of these subroutines are listed in Table 1.

All subroutines are called by the main program (*Main*). The computer code initially reads the input file *para.txt* and sets the required parameters in the subroutine ReadIn. Some significant matrices are pre-computed to improve the computational efficiency in the subroutine Storage. Next, the loop over the iteration number starts to operate. In each iteration, the magnetic fields are calculated layer by layer. For low magnetic susceptibility, having the first iterative cycle done is adequate enough to get accurate results. For high magnetic susceptibility, the total magnetic fields are calculated iteratively until the maximum number of iterations is reached or the required convergence accuracy is met. Finally, the magnetic gradient tensor is computed and then recorded in the output file *MagneticField_3D.dat*. The structure of the main program is shown in Figure 3.

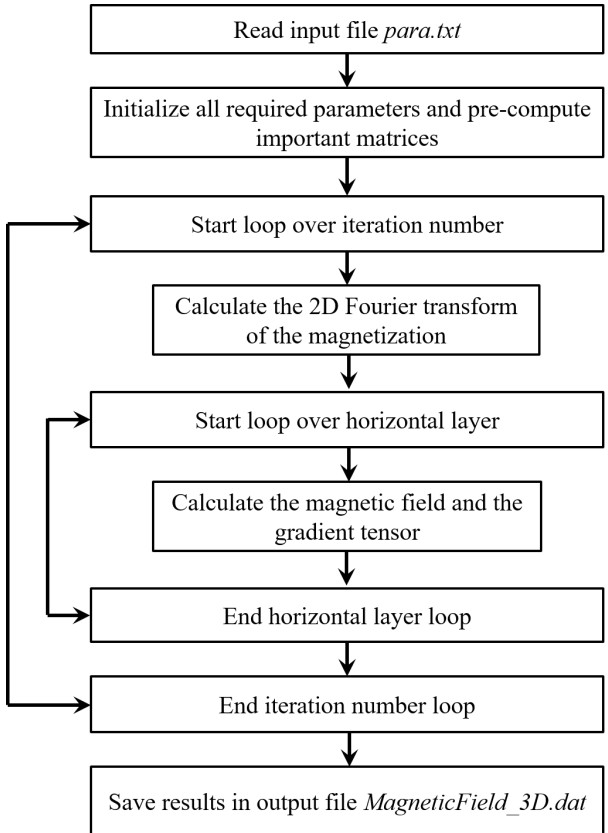

**Figure 3.** Basic structure of the main program.

**Table 1.** Subroutines for modeling the magnetic gradient tensor at high susceptibility.

| Main Program | Main | Perform the Iterative Procedure and Calculate the Magnetic Gradient Tensor |
|---|---|---|
| Major subroutines | ReadIn | Initialize all required parameters and read model parameters from the input file (*para.txt*). |
| | Storage | Pre-compute and save important matrices. |
| | Wavenumber | Generate wavenumbers for the 2D Gauss-FFT. |
| | Background | Set background fields. |
| | Model | Establish a model. |
| | GaussFFT_2D_Fast | Achieve forward and inverse 2D Gauss-FFT. |
| Major subroutines | Module | Include initial variable declarations. |
| | Coordinates | Generate rectangular prims. |
| | Output | Output the calculation results, including six gradient field components and three magnetic field components. |
| | DeOrAllocate | Allocate and deallocate parameters. |

### 3.2. Inputs and Outputs

The FORTRAN program requires a set of input parameters (see Table 2), such as the size of the computation space, the maximum iteration number and the strength and direction of the inducing magnetic field, etc. All these parameters are given in the input file *para.txt*. The magnetic susceptibility distribution of the studied magnetic bodies can be set freely in the subroutine Model. In the original version of the subroutine Model, we provide four types of magnetic models, including a sphere, a spherical shell, an ellipsoid and a two-dike model. One can modify the code in this subroutine to establish any expected

magnetic model with arbitrary susceptibility distributions. It is also noteworthy that the default values of the minimum x and y coordinates of the prism's center are set to be zero in the FORTRAN program, i.e., x0 = y0 = 0.

**Table 2.** The required input parameters in the file *para.txt*.

| | |
|---|---|
| Nx, Ny, Nz | Number of prismatic elements in the x, y and z directions. |
| z0 | The minimum z coordinate of geometric center of prisms (m). |
| x1, y1, z1 | The maximum x, y and z coordinates of geometric center of prisms (m). |
| NI | Maximum number of iterations |
| NG | Number of Gaussian nodes used in 2D Gauss-FFT (NG = 1, 2, 3 or 4) |
| Ang1 | Inclination angle (degree) |
| Ang2 | Declination angle (degree) |
| Bgr | Strength of the inducing field (nT) |

The output of the FORTRAN program is saved in a single file *MagneticField_3D.dat*. This file contains ten columns: $x$-coordinate, $y$-coordinate, $z$-coordinate, three magnetic field components ($B_x^a$, $B_y^a$, $B_z^a$, nT) and six magnetic gradient tensor components ($T_{xx}^a$, $T_{yy}^a$, $T_{zz}^a$, $T_{xy}^a$, $T_{xz}^a$, $T_{yz}^a$, nT/m).

## 4. Numerical Tests

In this section, the performance of the algorithm is evaluated and several numerical examples are presented to reveal the validity and applicability of the code.

### 4.1. Performance

To test the performance of the algorithm, we analyze its computational accuracy, convergence behavior and computational efficiency using several models (a spherical shell, a prolate ellipsoid and a sphere). The center of these magnetic bodies is located in the middle of the computational domain which extends from 0 km to 1 km in the $x-$, $y-$ and $z-$ directions. The strength, inclination and declination of the inducing magnetic field are 50,000 nT, 60° and 45°, respectively. Analytical solutions for the spherical shell model are provided in Appendix A. Because no analytical solution is available for the prolate ellipsoid, the reference gradient tensor of this model is computed by numerically taking derivatives of the magnetic field vector obtained from the routines of [36]. All tests are carried out on a personal computer with 2.3 GHz CPU and 12 GB RAM.

In order to verify the accuracy of the algorithm, we compare the six components of the magnetic gradient tensor calculated using the contraction integral method with the theoretical solutions. In the test, the computational domain is divided into $200 \times 200 \times 200$ regular prisms and the magnetic anomalies are a spherical shell with susceptibility of 50 SI and a prolate ellipsoid with susceptibility of 10 SI (see Figure 4).

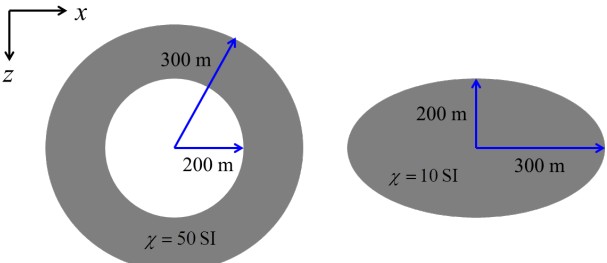

**Figure 4.** The magnetic spherical shell model with susceptibility of 50 SI and the prolate ellipsoid model with susceptibility of 10 SI used in the accuracy test.

Figures 5 and 6 show the numerical and analytical solutions of the two models and the differences between them. The statistical properties of the misfits for both models are listed in Table 3. As can be seen from Figure 5, the modeled results for the spherical shell have a perfect agreement with the analytical solutions. The rms error in this case is 0.082 for $T_{xx}^a$ and $T_{yy}^a$, 0.094 for $T_{xz}^a$ and $T_{yz}^a$, 0.046 for $T_{xy}^a$ and 0.142 for $T_{zz}^a$. The relative root mean square (rrms) errors in six magnetic gradient tensor components are all less than 0.5% (Table 3). For the prolate ellipsoid model, there is also a good agreement between the computed and reference solutions, but with a relatively larger rrms error (see Table 3). Part of such error is considered to be caused by the numerical derivation of the magnetic fields in order to obtain the reference magnetic gradient tensor.

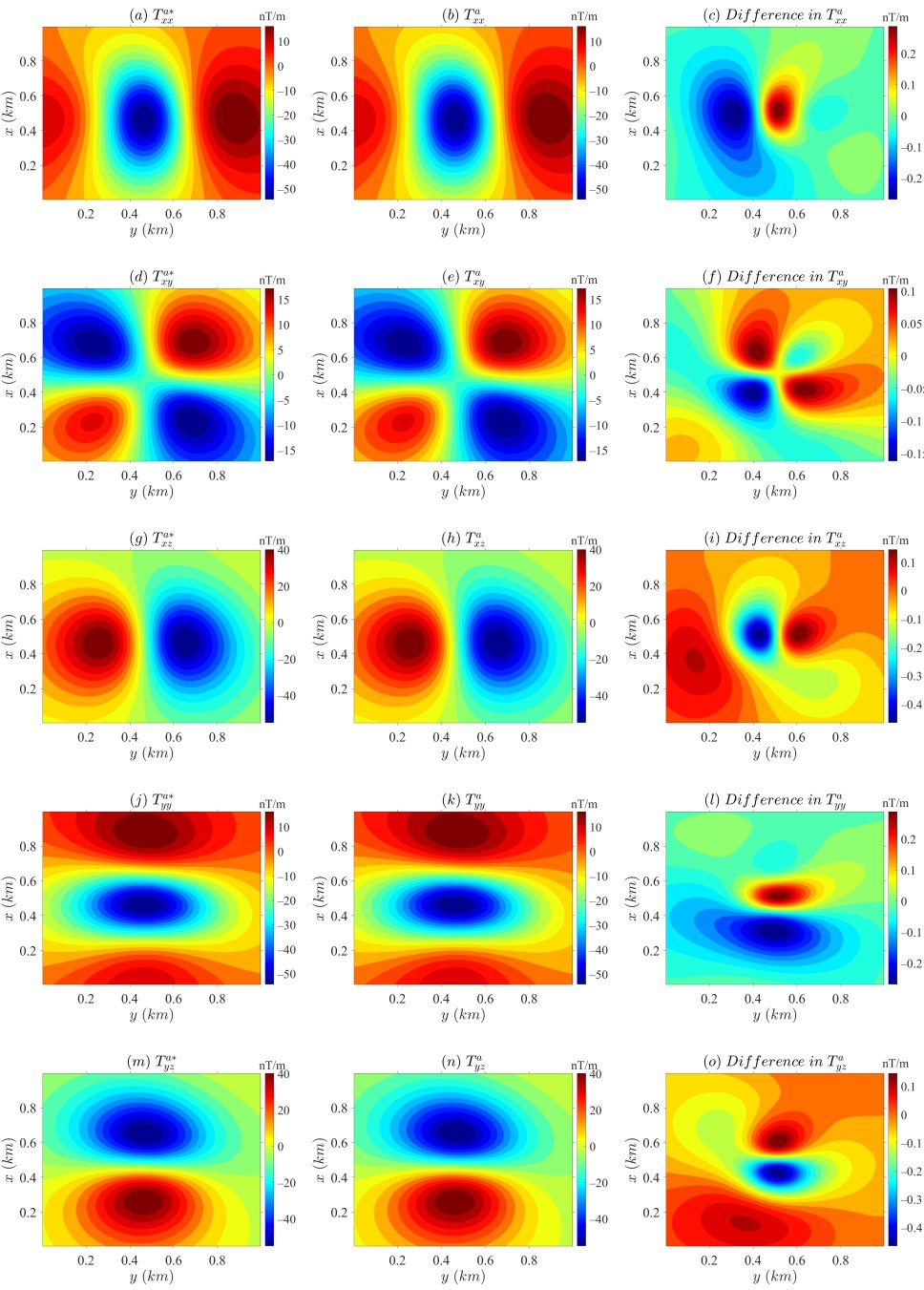

**Figure 5.** *Cont.*

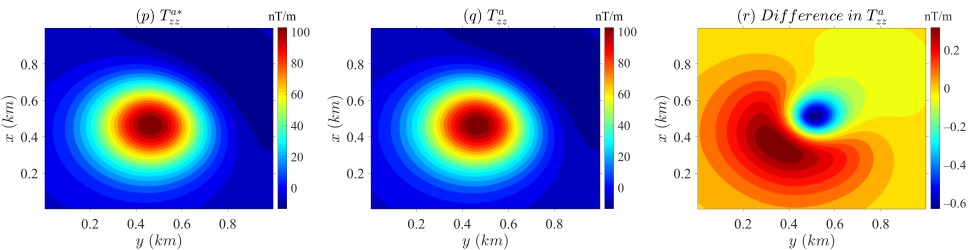

**Figure 5.** Surface anomalous magnetic gradient fields produced by a magnetic spherical shell with susceptibility of 50 SI. $T_{xx}^{a*}$, $T_{xy}^{a*}$, $T_{xz}^{a*}$, $T_{yy}^{a*}$, $T_{zz}^{a*}$, $T_{yz}^{a*}$ denote the analytical solutions, and $T_{xx}^{a}$, $T_{xy}^{a}$, $T_{xz}^{a}$, $T_{yy}^{a}$, $T_{zz}^{a}$, $T_{yz}^{a}$ denote the numerical results calculated using the proposed algorithm.

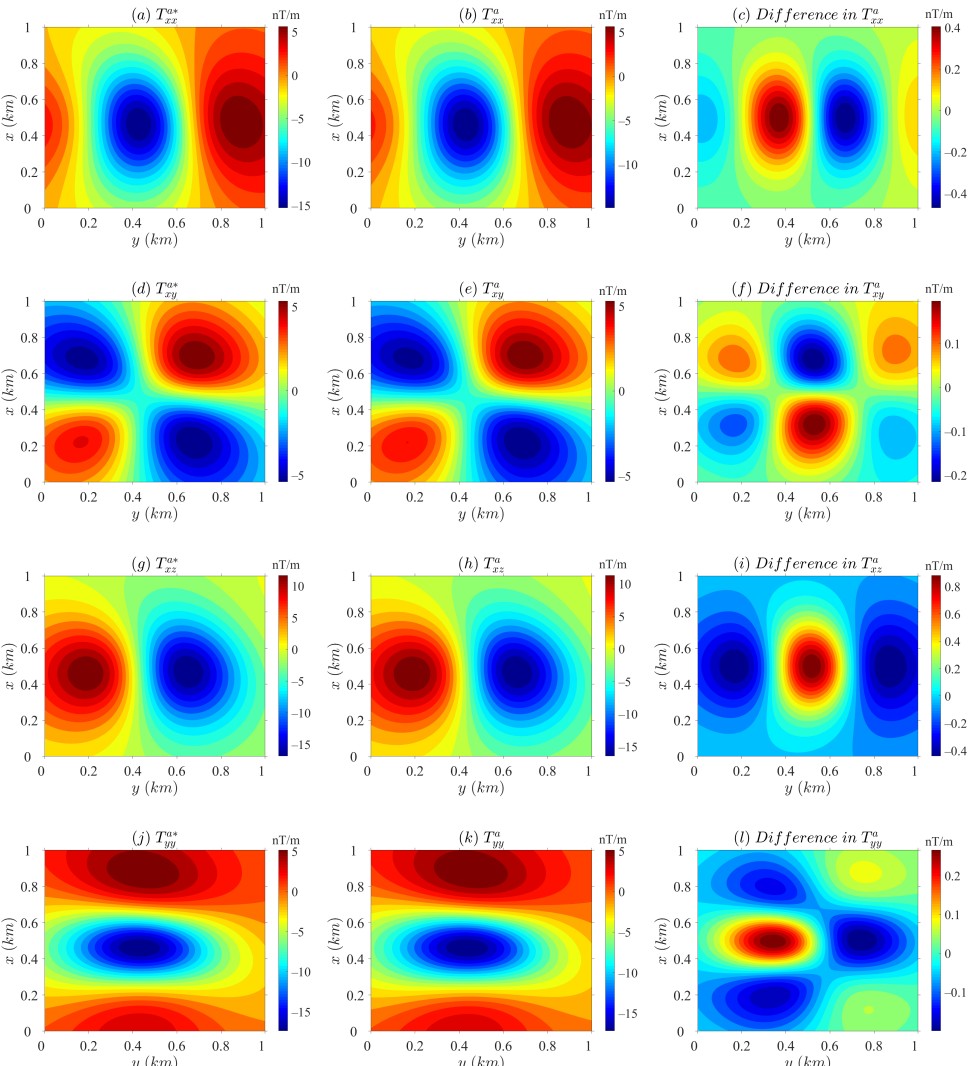

**Figure 6.** *Cont.*

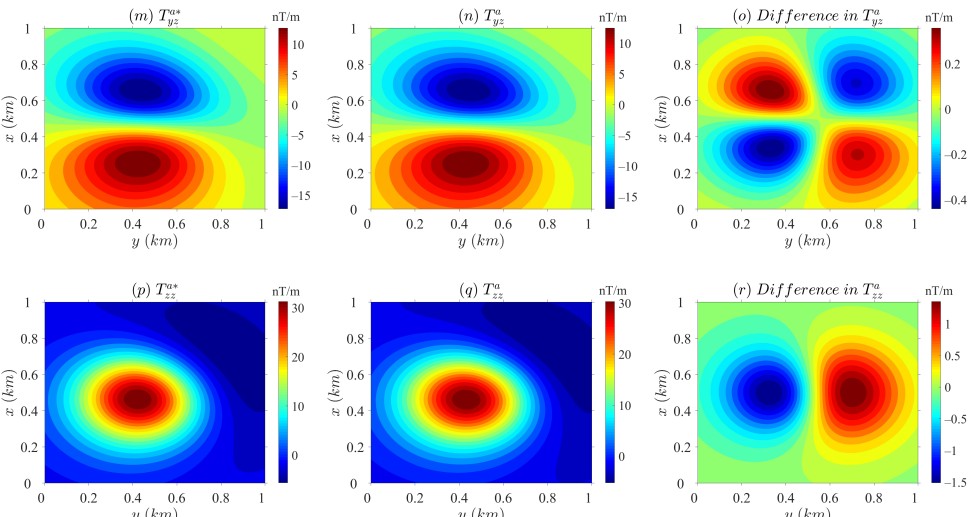

**Figure 6.** Surface anomalous magnetic gradient fields produced by a magnetic prolate ellipsoid with susceptibility of 10 SI. $T_{xx}^{a*}$, $T_{xy}^{a*}$, $T_{xz}^{a*}$, $T_{yy}^{a*}$, $T_{zz}^{a*}, T_{yz}^{a*}$ denote the analytical solutions, and $T_{xx}^{a}$, $T_{xy}^{a}$, $T_{xz}^{a}$, $T_{yy}^{a}$, $T_{zz}^{a}, T_{yz}^{a}$ denote the numerical results calculated using the proposed algorithm.

**Table 3.** Statistical parameters of the misfits for the spherical shell and the prolate ellipsoid.

| Gradient Fields | Min (nT) | | Max (nT) | | Rms | | Rrms * (%) | |
|:---:|:---:|:---:|:---:|:---:|:---:|:---:|:---:|:---:|
| | Shell | Ellipsoid | Shell | Ellipsoid | Shell | Ellipsoid | Shell | Ellipsoid |
| $T_{xx}^{a}$ | −54.188 | −14.931 | 19.733 | 6.691 | 0.082 | 0.158 | 0.455 | 2.812 |
| $T_{xy}^{a}$ | −17.159 | −5.358 | 18.884 | 5.818 | 0.046 | 0.079 | 0.467 | 2.533 |
| $T_{xz}^{a}$ | −54.908 | −16.482 | 44.953 | 12.703 | 0.094 | 0.276 | 0.431 | 4.031 |
| $T_{yy}^{a}$ | −54.188 | −17.190 | 19.733 | 6.210 | 0.082 | 0.092 | 0.455 | 1.534 |
| $T_{yz}^{a}$ | −54.908 | −16.983 | 44.953 | 13.923 | 0.094 | 0.177 | 0.431 | 2.459 |
| $T_{zz}^{a}$ | −13.582 | −5.235 | 108.377 | 32.115 | 0.142 | 0.565 | 0.457 | 5.538 |

* The relative rms error is defined as: $\mathrm{rrms}[T] = \mathrm{rms}[T − T^*]/\mathrm{rms}[T^*] \times 100\%$.

Next, we investigate the convergence behavior of the algorithm. To this end, we compute the convergence errors in three total magnetic field components based on a magnetic sphere model with different susceptibilities ranging from 1 SI to 1000 SI. Figure 7 shows the trend of the convergence error versus the number of iterations at different magnetic susceptibilities. We can see that the convergence behavior of the algorithm is significantly influenced by the magnitude of the magnetic susceptibility. At lower susceptibilities ($\chi \leq 10$ SI), the algorithm converges with a high speed. But when the magnetic susceptibility becomes larger ($\chi > 10$ SI), the convergence rate slows down. Although the convergence speed of the algorithm decreases with an increase of magnetic susceptibility, the algorithm exhibits a good and stable convergence behavior at a rather wide range of susceptibilities on the whole ($0 < \chi \leq 1000$ SI).

Now, we test the computation efficiency of the code. According to the theory of the algorithm mentioned in Section 2, the calculation of the magnetic gradient fields mainly consists of two parts. The first part is the computation of the frequency domain magnetic field at each wavenumber point ($k_x, k_y$). The efficiency of this part is mainly affected by the number of prismatic elements ($N_z$) used in the $z$ direction. The second part is the implementation of the 2D inverse Gauss-FFT applied to obtain the spatial domain wavefields. For this part, the calculation time is determined by the number of prismatic elements ($N_x \times N_y$) on the $x − y$ plane and the Gauss nodes (which is set to be 4 in our test) that one uses. Therefore, $N_z$ and $N_x \times N_y$ have different influence on the calculation speed.

To assess these factors, two cases are considered in this test. In case 1, we hold $N_h = 100$, where $N_h = N_x = N_y$, and set $N_z = 60, 80, 100, 150, 200, 250, 300$. In case 2, we hold $N_z = 100$ and set $N_h = 60, 80, 100, 150, 200, 250, 300$. Figure 8 shows the average computer time per iteration for calculating six magnetic gradient components and three magnetic field components in case 1 and case 2. As can be seen, when the number of the prismatic elements increases, the average computation time exhibits an approximately linear growth trend in both cases.

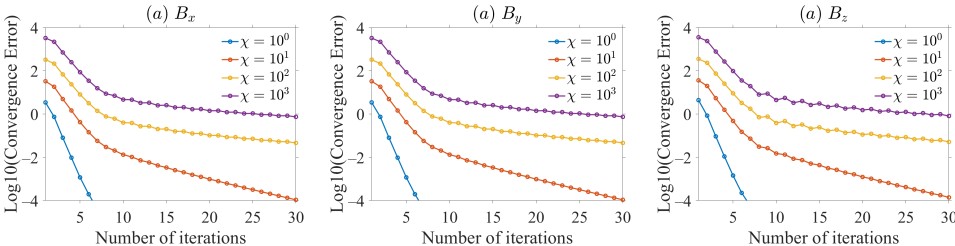

**Figure 7.** Convergence errors in $B_x$, $B_y$ and $B_z$ versus the number of iterations at different magnetic susceptibilities.

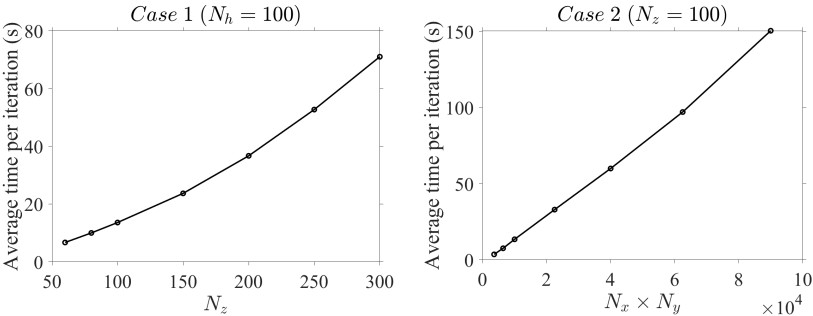

**Figure 8.** Average computer time per iteration for calculating six magnetic gradient components and three magnetic field components in case 1 and case 2.

### 4.2. Example

In this subsection, we present a synthetic example with a slightly more complicated geometry to further demonstrate the capability of the algorithm (Figure 9). This synthetic example is a two-dike model with a vertical dike striking east-west, and a stepped dike that strikes north-south and dips to the west at 45°. Both dikes spans a depth of 50–150 m and are assigned a high magnetic susceptibility value of 3 SI. The inducing field for this study has strength of 50,000 nT, inclination of 45° and declination of 45°. The whole computational space, with cells of 5 m cubed, extends from 0 m to 1000 m in the $x$ and $y$ directions and from 0 m to 250 m in the $z$ direction.

Figure 10 presents the surface total-field anomaly and the magnetic amplitude for the two-dike model with a magnetic susceptibility of 3 SI subject to a uniform inducing field. To verify the reliability of the computed results, we also display the magnetic data from a similar two-dike model calculated by [29] (see Figure 11). The geometric parameters of the west-dipping dike used in the work of [29] are slightly different from ours, but the values of other parameters are set the same. By comparing Figures 10 and 11, we can see that the magnetic fields obtained from both methods have consistent shape characteristics, which further validates our algorithm.

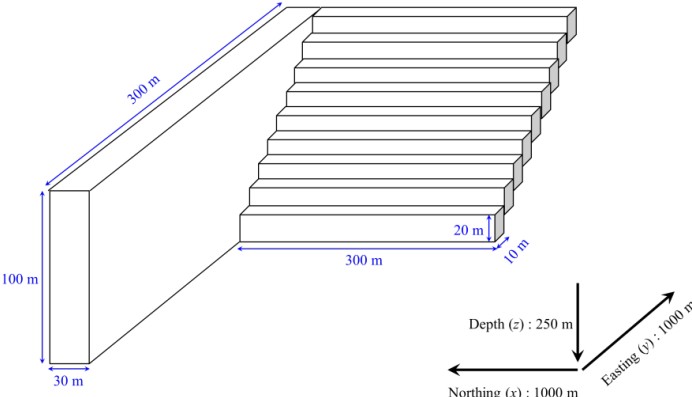

**Figure 9.** A synthetic two-dike model with a west-dipping dike striking north-south and a vertical dike striking east-west.

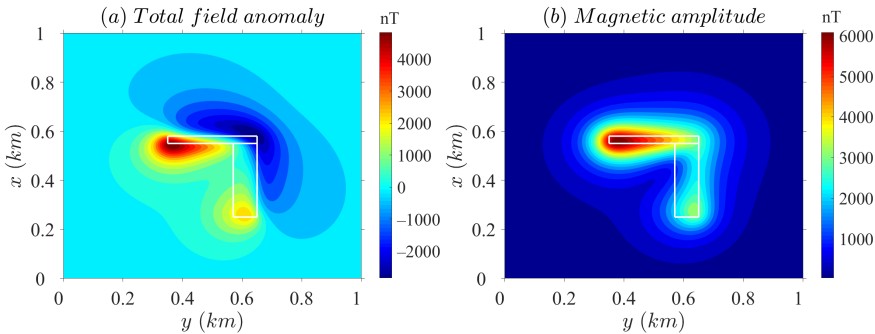

**Figure 10.** Surface total-field anomaly (**a**) and magnetic amplitude (**b**) due to the two-dike model with a magnetic susceptibility of 3 SI.

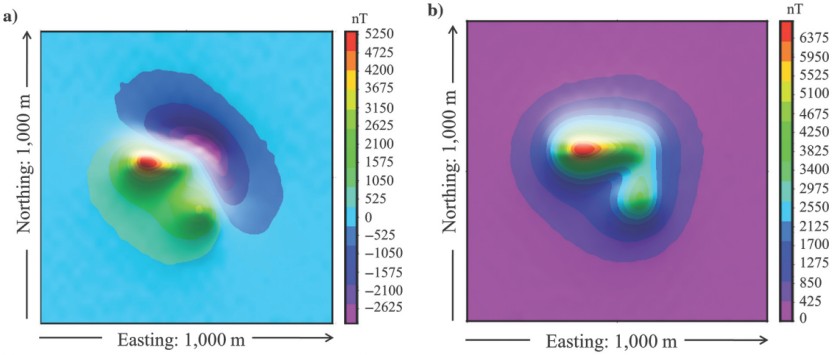

**Figure 11.** Total-field anomaly (**a**) and magnetic amplitude (**b**) from [29] for a similar two-dike model with a susceptibility of 3 SI.

In addition to the total-field anomaly, we also present the surface anomalous magnetic fields and gradient fields generated by the two-dike model in Figures 12 and 13, respectively. The white lines in these figures outline the projection of the magnetic bodies on the surface. As expected, the gradient tensor is superior to the magnetic field vector in enhancing the directional features of the magnetic anomaly.

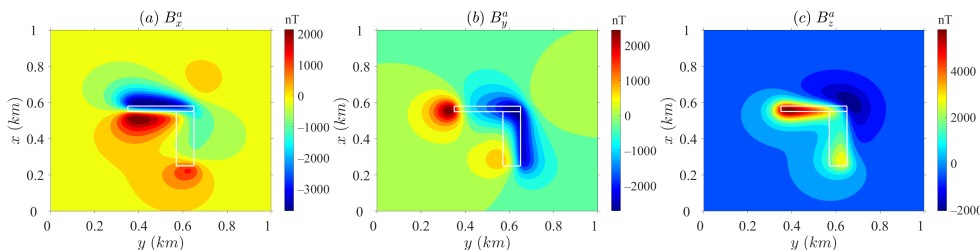

**Figure 12.** Surface anomalous magnetic fields due to the two-dike model with susceptibility of 3 SI.

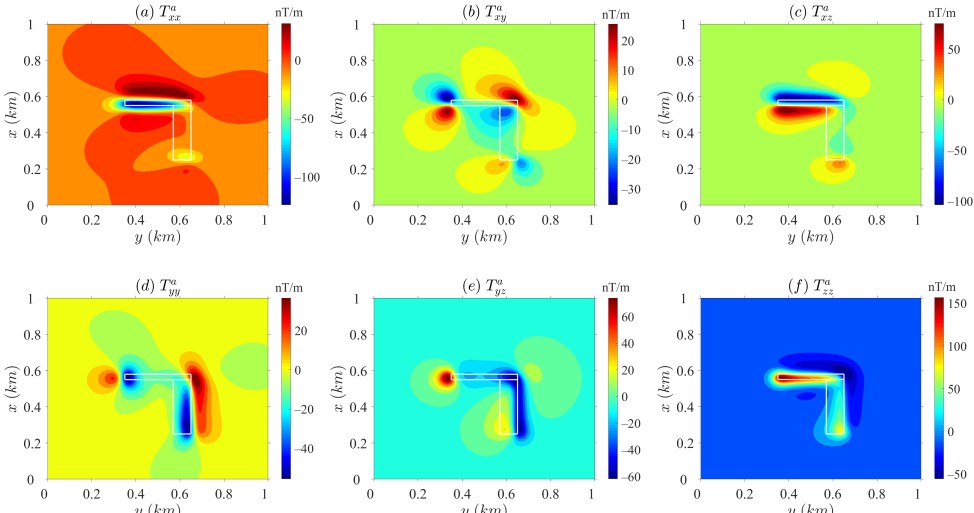

**Figure 13.** Surface anomalous magnetic gradient fields due to the two-dike model with susceptibility of 3 SI.

## 5. Conclusions

We have developed a computer program in FORTRAN programming language and provided a detailed description of the code for modeling magnetic gradient fields at high susceptibilities. The program generates multi-component fields produced by magnetic bodies of arbitrary shapes and high susceptibilities, including six components of the magnetic gradient tensor and three components of the magnetic field vector. A user friendly interface (i.e., input file) is also established to facilitate wide application of the FORTRAN program. Computational performance and capability of the scheme have been illustrated through several numerical examples (a spherical shell, a prolate ellipsoid, a sphere and a synthetic two-dike model). It is shown that the code performs well at a wide range of susceptibilities ($0 < \chi \leq 1000$ SI) and is particularly applicable for strongly magnetic bodies. This work, therefore, provides a significant open tool for modeling magnetic gradient fields at high susceptibility.

**Author Contributions:** Conceptualization, L.C.; methodology, L.C. and F.O.; software, F.O; formal analysis, F.O; resources, F.O; data curation, F.O; writing—original draft preparation, F.O; writing—review and editing, L.C.; visualization, L.C.; supervision, L.C.; project administration, L.C.; funding acquisition, L.C. All authors have read and agreed to the published version of the manuscript.

**Funding:** This research was funded by the Natural Science Foundation of Guangxi Province grant number 2020GXNSFDA238021.

**Institutional Review Board Statement:** Not applicable.

**Informed Consent Statement:** Not applicable.

**Data Availability Statement:** The name of the FORTRAN program (version 1) is 3D-MGTM-HS. This program is free and available at https://github.com/Yonfou/3D-MGTM-HS (accessed on 2

**Acknowledgments:** The authors are very grateful to the two anonymous reviewers for their helpful comments and constructive suggestions that helped to improve the paper.

**Conflicts of Interest:** The authors declare no conflict of interest.

## Appendix A. Analytical Solutions

The analytical solutions for anomalous magnetic gradient fields outside a magnetic spherical shell have the forms of

$$
\begin{aligned}
T_{xx}^{a} &= A_4\left(H_x^0\alpha_1\beta_1 + H_y^0\alpha_2\gamma_1 + H_z^0\alpha_3\gamma_1\right) \\
T_{xy}^{a} &= A_4\left(H_x^0\alpha_2\gamma_1 + H_y^0\alpha_1\gamma_2 - H_z^0\alpha_1\gamma_4\right) \\
T_{xz}^{a} &= A_4\left(H_x^0\alpha_3\gamma_1 - H_y^0\alpha_1\gamma_4 + H_z^0\alpha_1\gamma_3\right) \\
T_{yy}^{a} &= A_4\left(H_x^0\alpha_1\gamma_2 + H_y^0\alpha_2\beta_2 + H_z^0\alpha_3\gamma_2\right) \\
T_{yz}^{a} &= A_4\left(H_y^0\alpha_3\gamma_2 - H_x^0\alpha_1\gamma_4 + H_z^0\alpha_2\gamma_3\right) \\
T_{zz}^{a} &= A_4\left(H_x^0\alpha_1\gamma_3 + H_y^0\alpha_2\gamma_3 + H_z^0\alpha_3\beta_3\right)
\end{aligned}
\tag{A1}
$$

with

$$
A = \frac{\chi(2\chi + 3)\left(1 - r_i^{-3}r_o^3\right)}{2\chi^2 r_o^{-3} - (2\chi+3)(3+\chi)r_i^{-3}}
\tag{A2}
$$

and

$$
\begin{aligned}
&\alpha_1 = \frac{3(x-x_0)}{r^5}, \quad \alpha_2 = \frac{3(y-y_0)}{r^5}, \quad \alpha_3 = \frac{3(z-z_0)}{r^5} \\
&\beta_1 = 3 - \frac{5(x-x_0)^2}{r^2}, \quad \beta_2 = 3 - \frac{5(y-y_0)^2}{r^2}, \quad \beta_3 = 3 - \frac{5(z-z_0)^2}{r^2} \\
&\gamma_1 = \beta_1 - 2, \quad \gamma_2 = \beta_2 - 2, \quad \gamma_3 = \beta_3 - 2, \quad \gamma_4 = \frac{5(z-z_0)(y-y_0)}{r^2}
\end{aligned}
\tag{A3}
$$

where $r = \sqrt{(x-x_0)^2 + (y-y_0)^2 + (z-z_0)^2}$ is the distance between the observation point $(x, y, z)$ and the center of the magnetic spherical shell $(x_0, y_0, z_0)$, $r_i$ and $r_o$ denote the inner and outer radius of the spherical shell, respectively.

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
