# Peer review of "A FORTRAN Program to Model Magnetic Gradient Tensor at High Susceptibility Using Contraction Integral Equation Method"

_minerals, doi:10.3390/min11101129_

Round 1
Reviewer 1 Report
This is an implementation of known methods to calculate induced anomaly fields in FORTRAN. After checking the correctness of equation and examples, I do not have major comments. Here some minor points:
Line 116: replace "constant" with "homogeneous".
Line 117: It becomes clear few rows below, however I would add "frequency domain of the horizontal coordinates"
Eq. (12) and other equations: It is better to separate the components of a vector by comma, instead of space.
Line 119: Would sing not just be the sign function. Never heard of a the symbolic function in this context.
Fig.7: Specify that errors are in %.
Author Response
We greatly appreciate your timely comments and suggestions. The following are our point-by-point replies to the comments.
Line 116: replace "constant" with "homogeneous".
Answer: Accepted. The expression is changed to “The magnetization is assumed to be homogeneous in each prism.”
Line 117: It becomes clear few rows below, however I would add "frequency domain of the horizontal coordinates"
Answer: Accepted. The expression is changed to “We deal with Equation (8) in the frequency domain of the horizontal coordinates so that……”.
Eq. (12) and other equations: It is better to separate the components of a vector by comma, instead of space.
Answer: Accepted. Eq.(12) , (15) and Equation in Fig.2 are revised by separating the components of a vector by comma.
Line 119: Would sing not just be the sign function. Never heard of a the symbolic function in this context.
Answer: Our mistake. “the symbolic function” is changed to “the sign function” in the context.
Fig.7: Specify that errors are in %.
Answer: The root mean square difference, Eq. (21), is used to compute the convergence error in Fig. 7. Errors have the same dimension with magnetic field intensity. Thus there is no need “%” in the Figure. We notice that 100% in Eq. (21) is not proper. 100% in Eq. (21) is deleted in the new version.
Reviewer 2 Report
In this paper, the authors present a contraction integral equation method to simulate the gradient fields produced by 3-D magnetic bodies of arbitrary shapes and high susceptibilities. Recently (see ref [30]) the authors develop an efficient and accurate frequency-domain iterative method that can be used to simulate magnetic fields from magnetic bodies with arbitrary shapes in a wide range of magnetic susceptibilities (0 1000 SI). It is based on a contraction integral equation and can achieve fast convergence, however, for intermediate susceptibilities as shown in [30]. In the extreme case (1000 SI), an acceptable result was also obtained after sufficient iterative computation. In [30] it was shown that a further improvement in the numerical precision can be achieved by increasing the number of prismatic voxels, but it takes longer time... In the present study the authors explain that previous methods mainly focus on the calculation of the magnetic fields, and take no account of the gradient tensor. In this work the authors present an algorithm to compute the magnetic gradient fields produced by 3-D magnetic bodies of arbitrary shapes and high susceptibilities based on the contraction integral equation method developed by [30]. To facilitate application, the authors introduce a FORTRAN program to implement the algorithm. This program is intended for users who show interests in 3D magnetic modeling at high susceptibility. Therefore, the study is original and relevant and important for the readers of this Journal. Moreover, the authors present the program as free and available online.
However, I have minor remarks:
It would be good in the Introduction to explain in more details why the authors have chosen a Fortran programming language. Also, in the Conclusions: what would be recommendations for extremely high susceptibilities.
Author Response
Thank you for your suggestive remarks. We reply to your concern.
First is about the reason of choosing Fortran programming language. Fortran is always primary choice for numerical computation and scientific computation purpose in the community of computational geophysics. Using the same programming language benefits codes sharing. Most of our research work on numerical methods are codes using Fortran. We cannot tell at the moment if it is superior or inferior compared with other languages, such as C, Matlab, et al. from the perspective of implementation efficiency.
The next concern is how to deal with the case of extremely high susceptibility. The presented method is suited for the magnetic susceptibility varying from 0 SI to 1000 SI, which largely satisfies the requirement of geophysical exploration. As concerning about those beyond 1000 SI, from our experience, only applying fine subdivision is not enough. It converges quite slowly, or even not converge. Non-structure subdivision scheme and vector finite element method could be considered to combine with integral equation method for such extremely susceptibility cases. Further research work has to be done to verify that point.